# Genome-wide association study reveals a locus in *ADARB2* for complete freedom from headache in Danish Blood Donors
Isa Amalie Olofsson [1,2,3], Ragnar P. Kristjansson[1], Ida Callesen[1], Olafur Davidsson[1], Bendik Winsvold [4,5,6], Henrik Hjalgrim [7], Sisse R. Ostrowski [8,9], Christian Erikstrup [10,11], Mie Topholm Bruun [12], Ole Birger Pedersen [13], Kristoffer S. Burgdorf[8], Karina Banasik [14], Erik Sørensen[8], Christina Mikkelsen [8], Maria Didriksen [8], Khoa Manh Dinh [10], Susan Mikkelsen[10], International Headache Genetic Consortium*, DBDS Genomic Consortium*, Søren Brunak [14], Henrik Ullum[7], Mona Ameri Chalmer [1], Jes Olesen[1], Lisette J. A. Kogelman [1,2,15] & Thomas Folkmann Hansen [1,2,14,15] ✉

Headache disorders are the most common disorders of the nervous system. The lifetime prevalence of headache disorders show that some individuals never experience headache. The etiology of complete freedom from headache is not known. To assess genetic variants associated with complete freedom from headache, we performed a genome-wide association study of individuals who have never experienced a headache. We included 63,992 individuals (2,998 individuals with complete freedom from headache and 60,994 controls) from the Danish Blood Donor Study Genomic Cohort. Participants were included in two rounds, from 2015 to 2018 and in 2020. We discovered a genome-wide significant association, with the lead variant rs7904615[G] in *ADARB2* (EAF = 27%, OR = 1.20 [1.13–1.27], $p = 3.92 \times 10^{-9}$). The genomic locus was replicated in a non-overlapping cohort of 13,032 individuals (539 individuals with complete freedom from headache and 12,493 controls) from the Danish Blood Donor Study Genomic Cohort ($p < 0.05$, two-sided). Participants for the replication were included from 2015 to 2020. In conclusion, we show that complete freedom from headache has a genetic component, and we suggest that *ADARB2* is involved in complete freedom from headache. The genomic locus was specific for complete freedom from headache and was not associated with any primary headache disorders.

Headache disorders are the most common disorders of the nervous system and the leading cause of disability among young women[1,2]. The lifetime prevalence of headache is estimated to be 94% in Europe, with the highest prevalence of 96% reported in Denmark[3,4]. Concordantly, in a population-based study from the Danish Blood Donor Study we found that 4% of participants had never experienced a headache[5]. Understanding why these individuals never experience headache, may give insight into mechanisms that protect against headache.

Here, we call the phenotype of individuals who have never experience a headache, complete freedom from headache (CFH). Research on CFH is very limited, but we have earlier described a female to male ratio of 1:2.2 in CFH and shown that CFH is not associated with a higher socio-economic status or a healthier lifestyle compared to the general population[5]. Two studies have reported a decreased muscle tenderness in individuals with CFH[6,7]. This finding could not be confirmed in our recent study on pain sensitivity in men with CFH[8]. The nitric oxide signaling pathway has been implicated in primary headache disorders[9,10]. In a provocation study with nitric oxide in men, we found no difference in induced headache between individuals with CFH and controls[11]. Consequently, the nitric oxide pathway does not appear to play a role in headache protection in CFH. CFH has not been the focus of any previous genetic research, however, a large twin study of 11,199 twin pairs estimated a significant heritable component to having no experience of headache in the preceding year[12]. Whether a specific genetic component is associated with CFH remains unknown.

---

A full list of affiliations appears at the end of the paper. *Lists of authors and their affiliations appear at the end of the paper.
✉e-mail: Thomas.folkmann.hansen@regionh.dk

https://doi.org/10.1038/s42003-024-06299-y **Article**

We present a genome-wide association study comparing individuals with CFH to the general population, using 63,992 individuals (2998 individuals with CFH and 60,994 controls). We discovered a genome-wide significant locus at the *ADARB2* gene, which we replicated in a cohort of 13,032 individuals (539 individuals with CFH and 12,493 controls) and assessed its biological impact and association with other complex traits.

## Results

### Association analysis

After quality control 63,992 participants (2998 individuals with CFH and 60,994 controls) of North European ancestry and 11,283,815 single-nucleotide polymorphisms (SNPs) remained for association analysis.

We identified one genome-wide significant locus on chromosome 10, in the first intron of *ADARB2* encoding Adenosine Deaminase RNA Specific B2. The lead SNP was the intronic variant rs7904615[G] (effect allele frequency (EAF) = 27%, OR = 1.20[1.13–1.27], $p = 3.92 \times 10^{-9}$), Fig. 1. The SNP heritability for CFH was 3.71% (SE = 3.05, $p = 0.11$) on the liability scale. We had a statistical power of >80% to detect common variants (minor allele frequency=0.3) with small-to-moderate effect sizes (odds ratio (OR) > 1.2), and low-frequency variants (minor allele frequency=0.05) with moderate-to-large effect sizes (OR > 1.45).

We replicated the genetic risk locus of *ADARB2* in a non-overlapping cohort of 13,032 participants (539 individuals with CFH and 12,493 controls). The lead SNP in the genomic risk locus did not reach statistical significance ($p = 0.20$), however, the direction of its effect was replicated

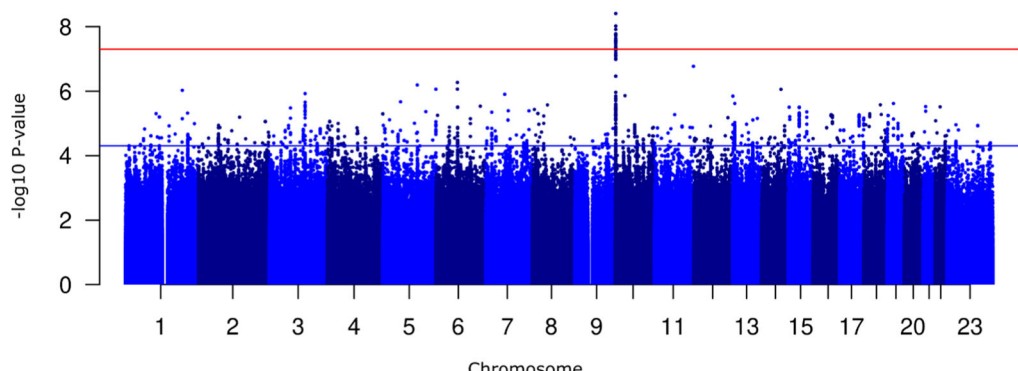

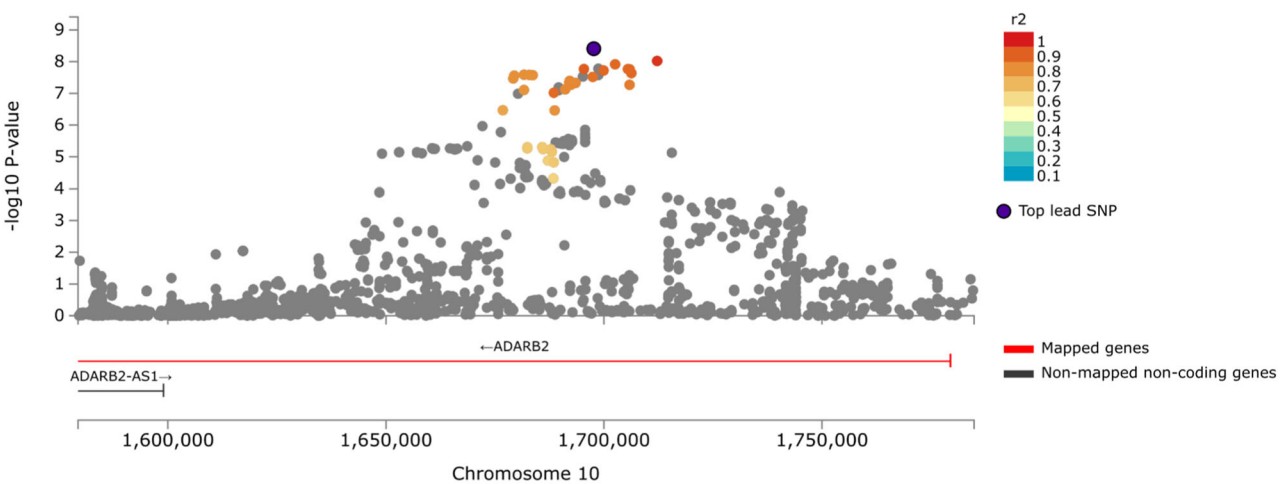

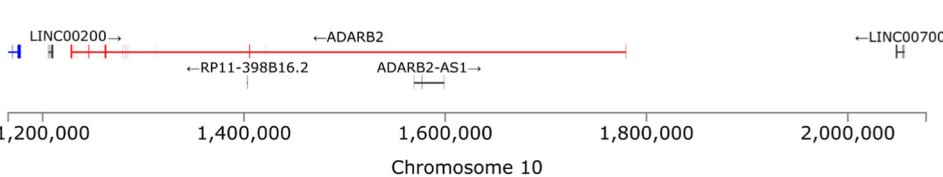

**Fig. 1 | The significant loci in *ADARB2* associated with complete freedom from headache (CFH).** Top panel: Manhattan plot of the GWAS of CFH. The red horizontal line denotes genome-wide significance ($p = 5 \times 10^{-8}$) and the blue horizontal line denotes suggestive association ($p = 5 \times 10^{-5}$). Mid panel: Regional locus zoom of the genome-wide significant risk locus, NCBI Build19. Each dot is colored by r² of linkage disequilibrium (LD) with purple-colored lead SNP. 2,998 cases and 60,994 controls were analyzed. Bottom panel: Regional plot of *ADARB2* and nearby genes, NCBI Build19.

 

https://doi.org/10.1038/s42003-024-06299-y **Article**

(OR = 1.19 [0.91–1.55]). Six out of seven genome-wide significant SNPs in the risk locus in linkage disequilibrium (LD) ($r^2 > 0.6$) with the lead SNP (rs7904615), were replicated ($p < 0.05$)(Table 1). In the meta-analysis of the discovery and replication cohorts, all variants were found to be genome-wide significant with no significant heterogeneity between the cohorts. This indicates robust associations with *ADARB2* across the combined cohorts (Table 1).

## Biological impact

All genome-wide significant SNPs mapped to the *ADARB2* gene, both based on positional and eQTL mapping. None of the *ADARB2* SNPs were previously associated with any of the >5,000 human traits in the PheWAS of the NHGRI-EBI GWAS Catalog[13]. RNA expression of *ADARB2* showed tissue enrichment in the brain with 2-8 times increased expression compared to the highest expression in non-brain tissue from The Human Protein Atlas[14], with the most pronounced expression in the spinal cord and the midbrain with nTPM (normalized protein-coding transcripts per million) of 31.9 and 20.1. Single-cell RNA expression showed enrichment of expression in inhibitory neurons and oligodendrocyte precursor cells with a tau specificity score of 0.87 in the Human Protein Atlas[14]. We found no data on tissue-specific expression of the gene product of *ADARB2* in the Human Protein Atlas[14]. No significant protein quantitative trait loci (pQTL) were reported for the lead variant in The Human Protein Atlas or EpiGraphDB[14,15]. A protein-protein interaction network for *ADARB2* showed a significant enrichment ($p = 1.93 \times 10^{-4}$) of biological processes involved in nucleobase metabolism($p = 3.7 \times 10^{-3}$), transfer-RNA editing ($p = 1.2 \times 10^{-3}$), and adenosine editing ($p = 2.2 \times 10^{-3}$)[16]. We found no drug-gene interactions for *ADARB2*.

## Polygenic architecture

Participants with CFH had a lower polygenic risk of migraine than controls (OR = 0.76 [0.75–0.80], $p = 3.67 \times 10^{-42}$). The lower polygenic risk of migraine was not due to participants with migraine among the controls ($N = 9273$), as shown by excluding participants with migraine from the analysis (OR = 0.82 [0.79–0.86], $p = 3.18 \times 10^{-24}$). Genetic correlation showed a negative genetic correlation with migraine ($r_g = -0.73$, $p = 4.65 \times 10^{-5}$).

## Discussion

Our analysis of 63,992 individuals identified a complete freedom from headache (CFH) risk locus in *ADARB2*. We replicate the genomic risk locus in a non-overlapping cohort, although the lead SNP did not reach statistical significance. Common variants accounted for 3.71% of the phenotypic variability in CFH.

Genome-wide significant variants were all located in the first intron of *ADARB2*. The gene encodes an RNA-editing enzyme expressed in the brain, primarily in inhibitory neurons[17,18]. *ADARB2* has been suggested to play a regulatory role in RNA editing and among the related pathways of *ADARB2* is ATP/ITP metabolism[19–21]. Results from protein-protein interactions of ADARB2 showed an overrepresentation of domains involved in RNA and DNA metabolism and editing.

Other variants in *ADARB2* have been associated with several brain disorders, including amyotrophic lateral sclerosis and Alzheimer's disease, longevity and different types of cancers[22–29]. *ADARB2* SNPs have been associated with migraine in one study on a small isolated island population[30]. However, the variants in *ADARB2* found in the island population were located more than 400,000 bp upstream of our lead SNP. Furthermore, the association between *ADARB2* and migraine was not replicated in a hypothesis-driven case-control study or a large GWAS meta-analysis of more than 100,000 migraine cases[31,32]. The lead SNP (rs7904615) from our analysis was not associated with migraine ($p = 0.85$), cluster headache ($p = 0.12$) or tension-type headache ($p = 0.39$) in the respective GWAS summary statistics (unpublished tension-type GWAS summary statistics provided by the Norwegian HUNT study)[32–34].

**Table 1 | Replication of the genome-wide significant risk locus**

| SNP | Chr:Pos | Allele (effect/other) | EAF | Impute quality | Discovery OR [95% CI] | Discovery p-value | Replication OR [95% CI] | Replication p-value | Meta OR [95% CI] | Meta p-value | p-het | I² |
|---|---|---|---|---|---|---|---|---|---|---|---|---|
| rs4880903 | 10:1653224 | A/G | 0.25 | 0.998 | 1.19 [1.12-1.27] | $1.72 \times 10^{-8}$ | 1.50 [1.13-2.00] | $5.40 \times 10^{-3}$ | 1.20 [1.13-1.28] | $1.11 \times 10^{-9}$ | 0.12 | 58.6 |
| rs7904615 | 10:1655482 | G/A | 0.27 | 0.994 | 1.19 [1.13-1.27] | $3.92 \times 10^{-9}$ | 1.19 [0.91-1.55] | 0.20 | 1.20 [1.13-1.27] | $1.68 \times 10^{-9}$ | 0.97 | 0.00 |
| - | 10:1656531 | -/G | 0.25 | 0.992 | 1.19 [1.12-1.27] | $1.66 \times 10^{-8}$ | 1.47 [1.10-1.96] | $8.44 \times 10^{-3}$ | 1.20 [1.13-1.28] | $1.28 \times 10^{-9}$ | 0.16 | 48.8 |
| rs7071633 | 10:1657739 | A/G | 0.25 | 0.994 | 1.19 [1.12-1.27] | $1.91 \times 10^{-8}$ | 1.45 [0.92-1.92] | $1.10 \times 10^{-2}$ | 1.20 [1.13-1.28] | $1.65 \times 10^{-9}$ | 0.19 | 41.5 |
| rs11250714 | 10:1660387 | T/C | 0.25 | 0.996 | 1.19 [1.12-1.27] | $1.22 \times 10^{-8}$ | 1.50 [1.13-2.01] | $5.52 \times 10^{-3}$ | 1.21 [1.14-1.28] | $7.82 \times 10^{-10}$ | 0.12 | 57.6 |
| rs4880904 | 10:1663314 | G/A | 0.25 | 0.996 | 1.19 [1.12-1.27] | $1.72 \times 10^{-8}$ | 1.48 [1.11-1.96] | $7.64 \times 10^{-3}$ | 1.20 [1.13-1.28] | $1.28 \times 10^{-9}$ | 0.15 | 51.4 |
| rs4880524 | 10:1663712 | A/T | 0.25 | 0.996 | 1.19 [1.12-1.27] | $1.75 \times 10^{-8}$ | 1.47 [1.11-1.96] | $7.76 \times 10^{-3}$ | 1.20 [1.13-1.28] | $1.31 \times 10^{-9}$ | 0.15 | 51.1 |
| rs3750685 | 10:1670015 | T/C | 0.27 | 0.992 | 1.19 [1.12-1.26] | $9.53 \times 10^{-9}$ | 1.20 [0.92-1.56] | 0.19 | 1.19 [1.12-1.26] | $3.93 \times 10^{-9}$ | 0.98 | 0.00 |

Odds ratios and p-values for the lead SNP and seven genome-wide significant SNPs in LD with it, from the discovery analysis, the replication analysis and the meta-analysis of both cohorts. *SNP* single nucleotide polymorphism, *Chr* chromosome, *Pos* position in NCBI Build38, *EAF* effect allele frequency, *Impute quality* statistical quality of genotype imputation, *OR* odds ratio, *CI* confidence interval, *p-het* p-value of the heterogeneity, *I²* percentage of variance attributable to heterogeneity.

 3

**Table 2 | Demographic description of the discovery and replication cohorts**

| | Discovery cohort (*N* = 63,992) | | | Replication cohort (*N* = 13,032) | | | *p*-value[a] |
|---|---|---|---|---|---|---|---|
| | Cases | Controls | All | Cases | Controls | All | |
| Women, *N* (%) | 1,032 (34.4%) | 31,196 (51.1%) | 32,201 (50.3%) | 161 (29.9%) | 6,540 (52.3%) | 6,709 (51.5%) | 0.028 |
| Age, mean (SD) | 51.8 (14.9) | 46.1 (13.9) | 46.3 (14.0) | 44.4 (14.2) | 39.4 (13.0) | 39.6 (13.1) | <0.001 |

Difference in sex distribution tested with a chi-square test and age tested with a two-sided *t*-test.
*Age* age in years, *SD* standard deviation.
[a]*p*-value comparing the total discovery cohort to the total replication cohort.

PheWAS analyses of *ADARB2* significant SNPs showed no association with other traits reported in the NHGRI-EBI GWAS Catalog[13]. In the GWAS Catalog *ADARB2* has been associated with 60 different traits[13]. Among the traits with the lowest *p*-value were height (GCST90245848), age of menarche(GCST007078), acute myeloid lymphoma(GCST008413), onset of male puberty(GCST90012088), depression (GCST007342) and creatinine levels (GCST90019502)[13]. Additional functional assessments of ADARB2, including effects of intronic variants on gene expression, are needed.

The polygenic risk score for migraine was lower in individuals with CFH than in controls. Importantly, the exclusion of participants with migraine from our analyses did not affect our results. CFH and migraine showed a significant negative genetic correlation. We do not expect that the negative genetic correlation is driven by individuals with CFH being part of the controls in migraine GWAS, as the prevalence of CFH is only 4.1%. The low polygenic risk score for migraine, together with the negative genetic correlation with migraine, might suggest a shared biology or indicate the existence of a biological continuum of susceptibility to headache. We speculate if *ADARB2* could affect susceptibility to headache by decreasing the individual headache threshold. This notion is supported by our clinical study where men with CFH experienced headache when provoked with nitric oxide, a strong headache trigger[11].

In conclusion, we present a GWAS on CFH and show that there is a genetic component to never having experienced headache. Our results suggest the involvement of *ADARB2*, a locus specific to CFH and independent of headache disorders. Further studies are needed before *ADARB2* can be proven as a gene contributing to protection from headache.

## Methods
### Participants
Participants were part of the Danish Blood Donor Study (DBDS)[35]. The DBDS is a nationwide population-based biobank and research platform that utilizes the existing infrastructure in the Danish blood banks for the prospective cohort. Blood donors must be healthy and comply with strict health criteria to donate blood. Upon enrollment, participants completed a comprehensive questionnaire and a whole blood sample was collected for genotyping. Included participants could consent to recontact with questionnaires or invitations to further research studies. To date, more than 150,000 blood donors aged 18-70 years have been included in the DBDS[35]. Genotyping has been performed among 114,000 DBDS participants forming the DBDS Genomic Cohort[36].

### Ethics
All participants provided written informed consent. All ethical regulations relevant to human research participants were followed. The DBDS dataset was approved by The Scientific Ethical Committee of Central Denmark (1-10-72-95-13) and of Zealand Region (SJ-740). The DBDS dataset was approved by the Danish Data Protection Agency (P-2019-99). GWAS studies in DBDS were approved by the National Ethical Committee (NVK-1700407).

**Discovery cohort**. The discovery cohort included 63,992 Danish adults (2998 individuals with CFH and 60,994 controls) of North European ancestry originating from the DBDS Genomic Cohort. CFH was defined based on the question: Do you believe that you never ever in your whole life have had a headache?. The question had a yes/no answer. All who answered yes were classified as individuals with CFH and all who answered no were classified as controls. The question was given to DBDS participants from November 2015 to March 2018 upon inclusion in the DBDS. From May 2020 to August 2020 a questionnaire with the question on CFH were sent to participants from the DBDS by recontact through secure electronic mail. If participants had answered the question on CFH in both rounds of the questionnaire, the newest answer where chosen. Demographic description of the discovery cohort, see Table 2 and Supplementary Table 1. There were more men among CFH cases than controls, with OR = 1.99 [1.85−2.15], *p* = 2.44 × 10$^{-71}$.

**Replication cohort**. The replication cohort included 13,032 Danish adults (539 individuals with CFH and 12,493 controls) of North European ancestry from the DBDS Genomic Cohort. Participants were included from November 2015 to March 2018 and from May 2018 to November 2020 when included in the DBDS as well as from May 2020 to August 2020 through recontact by electronic mail. There was no overlap with the discovery cohort. For participants included from November 2015 to March 2018 and from May 2020 to August 2020 CFH was defined based on the question: do you believe that you never ever in your whole life have had a headache? Participants were classified with CFH if they answered yes and as controls if they answered no. Participants included from May 2018 to November 2020 had answered the question: Have you experienced headache? Answers were on an ordinal scale: Never ever, rare, yearly, monthly, weekly or daily. Participants were classified with CFH if they answered never ever and as controls if they had answered any of the remaining five options. Demographic description of the replication cohort, see Table 2 and Supplementary Table 1.

### Genotyping and quality control
All genotype data, for both the discovery cohort and the replication cohort, was processed simultaneously for genotype calling, quality control and imputation. DNA was purified from whole blood and stored at −20 °C. The samples were genotyped using the Infinium Global Screening Array on the Illumina® genotyping platform by deCODE Genetics, Iceland[36]. The arrays cover more than 650,000 common genetic markers. Genotyping data were imputed using a reference panel consisting of 8000 Danish whole genome sequences, UK 1 KG phase 3 and HapMap to predict non-genotyped single nucleotide polymorphisms (SNPs) with minor allele frequency >1%[37,38]. Standard quality control was performed assessing individuals with sex discordance, missingness >0.03, heterozygosity of ±3 standard deviations (SD) from the mean and ancestry outliers defined as ± 5 SD from the mean, resulting in the exclusion of 219 individuals. Furthermore, SNPs with a missingness >0.02, Hardy Weinberg equilibrium *p* < 1 × 10$^{-10}$ and minor allele frequency <0.01 were excluded. Sequence variants were mapped to NCBI Build38[39].

## Statistics and reproducibility

Genome-wide association analysis was performed using a generalized linear mixed model to examine the association between each SNP and CFH. We assumed an additive genetic model adjusting for sex, age, age², the first ten genotype-derived principal components and kinship, using fastGWA-GLMM from GCTA v1.93 (github.com/jianyangqt/gcta)[40]. Sex and age were included in the model as pain sensitivity is influenced by both sex and age[41]. Age² was included due to the non-normal age distribution of the cohort. Plot of variance explained by principal component 1-10, see Supplementary Fig. 1. An independent significant SNP was defined as a one that reached genome-wide significance ($p < 5 \times 10^{-8}$) and was independent from other genome-wide significant SNPs with regard to linkage disequilibrium (LD) ($r^2 < 0.1$). Genome-wide significant SNPs were merged into a single genomic risk locus if they were located less than 250 kb apart. A genomic risk locus was defined as a region containing a genome-wide significant SNP and the surrounding SNPs in LD with the genome-wide significant SNP ($r^2 > 0.6$).

Power calculation was performed with a prevalence of 4.1%, a sample size of 63,992 (2,998 individuals with CFH and 60,994 controls) and a $p$-value threshold of $p < 5 \times 10^{-8}$. Power calculation was performed with the GAS Power Calculator (csg.sph.umich.edu/abecasis/cats/gas_power_calculator/).

Statistically significant genomic risk loci were replicated using a non-overlapping cohort from the DBDS Genomic Cohort with a sample size of 13,032 participants (539 individuals with CFH and 12,493 controls). Replication association analysis was performed adjusting for sex, age, age² and the first ten principal components using PLINK v.1.90[42]. Statistical significance was defined as $p < 0.05$ (two-sided).

Results from the two study cohorts were combined in a meta-analysis using the Mantel-Haenszel model[43]. The cohorts were allowed to have different population frequencies for alleles and genotypes but were assumed to have similar odds ratios. Heterogeneity was tested by comparing the null hypothesis of the effect being the same in all populations, to the alternative hypothesis of each population having a different effect using a likelihood ratio test. Statistical significance of heterogeneity was defined as $p < 0.05$.

## Biological impact of associated loci

Annotation of genome-wide significant loci, tissue expression analysis and MAGMA gene-set enrichment analysis were performed using FUMA v1.3.7[44,45]. Tissue expression analysis was performed using GTEx v.8 and BrainSpan data[46,47]. Gene set enrichment analysis was performed on genes identified by positional or expression quantitative trait loci (eQTL) mapping. Protein quantitative trait loci (pQTL) were assessed using The Human Protein Atlas and EpiGraphDB[14,15]. Protein-protein interaction networks of the mapped genes were examined with the STRING database[16]. A significant protein-protein interaction network was defined as $p < 0.05$. Phenome-wide association analyses (PheWAS) were conducted on genome-wide significant loci using >5000 human traits from the NHGRI-EBI GWAS Catalog[13]. Significant associations were defined as $p < 0.05$ after Bonferroni correction. Drug-gene interactions were performed on annotated genes using the Drug Gene Interaction Database (DGIdb)[48].

## Heritability and genetic correlations

SNP heritability was estimated using restricted maximum likelihood analysis adjusted for sex, age, age², principal component 1–10, and kinship using GCTA v1.93. The prevalence of CFH was set to 4.1%[5].

We estimated genetic correlation ($r_g$) between CFH and migraine using LD Score Regression (LDSC)[49]. We used the summary statistics from the most recent migraine meta-GWAS excluding Danish individuals and the 23andMe cohort[32]. Pre-calculated LD scores based on European individuals from the 1000 Genomes project were downloaded from alkesgroup.broadinstitute.org/LDSCORE/. For migraine the population prevalence was set at 15% and the sample prevalence (i.e. prevalence in migraine meta-GWAS) was set at 8%[32,50]. For CFH both the population prevalence and the sample prevalence (i.e. prevalence in the discovery cohort) were set at 4%[5]. Statistical significance was set at $p < 0.05$.

## Polygenic risk score

The polygenic risk score for migraine was calculated for the entire discovery cohort using LDpred2[51]. The polygenic risk score was based on summary statistics from the most recent migraine meta-GWAS excluding the Danish cohort and the 23andMe cohort[32]. A logistic regression model adjusted for sex, age, age², and principal component 1-5 was used to predict CFH. To ensure that the signal was not driven by migraine, we repeated the test excluding participants with migraine from the control group ($N_{migraine} = 9273$). Participants with migraine from the discovery cohort were identified using a validated self-reported migraine questionnaire[52].

## Reporting summary

Further information on research design is available in the Nature Portfolio Reporting Summary linked to this article.

## Data availability

The CFH GWAS summary statistics are available at Figshare.com with https://doi.org/10.6084/m9.figshare.25575204. For information on further access to data included in the analysis, please contact Thomas Folkmann Hansen (Thomas.Folkmann.Hansen@regionh.dk).

## Code availability

Variants in the Danish Blood Donor Genomic Cohort were imputed using software developed at deCODE genetics, Iceland. A generalized linear mixed model implemented by fastGWA-GLMM from GCTA v1.93 (github.com/jianyangqt/gcta) was used to test for association between sequence variants and CFH. We used publicly available software (URLs listed below) in conjunction with the above-described algorithm: The Drug Gene Interaction Database, dgidb.org/ EpiGraphDB, epigraphdb.org/ FUMA v1.3.7, fuma.ctglab.nl/ The Human Protein Atlas, proteinatlas.org/ LD Score Regression (LDSC), github.com/bulik/ldsc LDpred2, privefl.github.io/bigsnpr/articles/LDpred2 NHGRI-EBI GWAS Catalog, ebi.ac.uk/gwas/ STRING database, string-db.org/

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

## Acknowledgements

We thank the Danish blood donors for their valuable participation in the Danish Blood Donor Study as well as the staff at the blood centers for making this study possible. The work was supported by the Independent Research Fund Denmark (9039-00067B to T.F.H.), Candy Foundation (CEHEAD to J.O.), the Novo Nordisk Foundation (NNF20SA0064340 to

I.A.O.), by the Danish National Research Foundation (DNRF115) and South-Eastern Norway Regional Health Authority (2020034 and 2023089 to B.S.W.). The funding bodies had no role in the design of the study or in the analysis and interpretation of the data. We thank deCODE genetics, Reykjavik, Iceland for genotyping of the DBDS Genomic Cohort.

## Author contributions

Concept and design: I.A.O., R.P.K., J.O., T.F.H., and L.K. Data acquisition: B.W., H.H., S.R.O., C.E., M.T.B., O.B.P., K.S.B., K.B., E.S., C.M., M.D., K.M.D., S.M., S.B., H.U., M.A.C., and T.F.H., International Headache Genetic Consortium, DBDS Genomic Consortium. Analysis or interpretation of data: I.A.O., R.P.K., I.C., O.D., T.F.H., and L.K. Drafting of the manuscript: I.A.O. Critical revision of the manuscript: I.A.O., R.P.K., I.C., O.D., B.W., H.H., S.R.O., C.E., M.T.B., O.B.P., K.S.B., K.B., E.S., C.M., M.D., K.M.D., S.M., S.B., H.U., M.A.C., J.O., L.K., and T.F.H.

## Competing interests

The authors declare no competing interests.

## Additional information

[1]Danish Headache Center, Department of Neurology, Rigshospitalet, Copenhagen University Hospital, Glostrup, Denmark. [2]NeuroGenomic, Translational Research Center, Rigshospitalet, Copenhagen University Hospital, Glostrup, Denmark. [3]Department of Cellular and Molecular Medicine, Faculty of Health and Medical Sciences, University of Copenhagen, Copenhagen, Denmark. [4]Department of Research and Innovation, Division of Clinical Neuroscience, Oslo University Hospital, Oslo, Norway. [5]K.G. Jebsen Center for Genetic Epidemiology, Department of Public Health and Nursing, NTNU, Norwegian University of Science and Technology, Trondheim, Norway. [6]Department of Neurology, Oslo University Hospital, Oslo, Norway. [7]Statens Serum Institut, Copenhagen, Denmark. [8]Department of Clinical Immunology, Copenhagen University Hospital, Rigshospitalet, Copenhagen, Denmark. [9]Department of Clinical Medicine, Faculty of Heath and Medical Sciences, University of Copenhagen, Copenhagen, Denmark. [10]Department of Clinical Immunology, Aarhus University Hospital, Aarhus, Denmark. [11]Department of Clinical Medicine, Aarhus University, Aarhus, Denmark. [12]Department of Clinical Immunology, Odense University Hospital, Odense, Denmark. [13]Department of Clinical Immunology, Zealand University Hospital, Køge, Denmark. [14]Translational Disease Systems Biology, Novo Nordisk Foundation Center for Protein Research, University of Copenhagen, Copenhagen, Denmark. [15]These authors jointly supervised this work: Lisette J. A. Kogelman, Thomas Folkmann Hansen. ✉e-mail: Thomas.folkmann.hansen@regionh.dk

## International Headache Genetic Consortium

**Thomas Folkmann Hansen** ⓘ[1,2,14,15]✉, **Lisette J. A. Kogelman** ⓘ[1,2,15], **Mona Ameri Chalmer** ⓘ[1] & **Bendik Winsvold**[4,5,6]

## DBDS Genomic Consortium

**Thomas Folkmann Hansen** ⓘ[1,2,14,15]✉, **Henrik Hjalgrim**[7], **Sisse R. Ostrowski**[8,9], **Christian Erikstrup**[10,11], **Mie Topholm Bruun**[12], **Christina Mikkelsen**[8], **Maria Didriksen**[8], **Khoa Manh Dinh**[10], **Susan Mikkelsen**[10], **Ole Birger Pedersen**[13], **Kristoffer S. Burgdorf**[8], **Karina Banasik** ⓘ[14], **Søren Brunak** ⓘ[14] & **Henrik Ullum**[7]

A full list of members and their affiliations appears in the Supplementary Information.

