## [Peer Review File · Communications Biology]

Reviewers' comments:

Reviewer #1 (Remarks to the Author):

Authors describe their discovery of a protective genetic impact on headache using genome wide association study and the Danish Blood Donor Study Genomic Cohort data.

This is an interesting finding observed in the big sample size of Danish population and then replicated in a different cohort although in the Danish population as well.

However here are questions and recommendations that can clarify and improve the manuscript.

1. The title of the paper should reflect usage of the Danish Blood Donor Study Genomic Cohort data.
2. I recommend to remove the second sentence from the abstract "Yet, 4% of the Danish population have never experienced headache". This statement requires a reference and usually an abstract doesn't contain cited publications.
3. Similarly, the sentence "ADARB2 is primarily expressed in the brain, but the function of the gene is poorly understood" can be removed from the abstract.
4. On the contrary, it would be valuable to include in the abstract different time frames for data collection for the discovery cohort (2015-2018) and for the replication cohort (2018-2020) if my understanding is correct.
5. I recommend to add information to the Introduction that 4% prevalence of headache free participants was previously published by the same group of authors. Additionally, ratio for complete freedom from headache (CFH) for sex, socio-economic status was also self-cited. Please, tell readers if protective genetic impact was reported by other researchers since other references in the Introduction describe genetic findings associated with headaches.
6. Why were participants contacted twice: in 2015-2018 and then in May 2020 – August 2020? Were you able to contact all 63,992 participants in 2020?
7. Lines 76-77 tell us that genotyping and imputation was described in the paper submitted for publication in 2018. Please, clarify if you performed QC (and how) for participants from 2018-2020 years.
8. Why did you include 10 PCs in the regression models? A scree plot can be interesting to see (maybe in the supplemental materials). What was your rationale to include sex, age, and age squared as co-variables in the regression models? A summary of base models for discovery and replication cohorts (coefficients and p-values) should be included in the main text or supplemental materials.
9. Demographic description of participants (sex, age) has to be reported for both cohorts.
10. What formula or software was used for power calculation? Results for Power calculation should be moved to the Results section.
11. Please make a reference for 23andMe data, line 137.
12. Please, explain differences in population prevalence for migraine (15%) and sample prevalence (8%). Please, provide references, line 140.
13. Please, put the reference for LDpred2 next to it, line 145. References at the end of sentences (30, 31 - line 146) are confusing. If you want to be consistent and keep references at the end of sentences, then split the first sentence in the section about Polygenic risk score (lines 144-146).
14. Why only 5 PCs were used for the polygenic risk score regression model?
15. How many SNPs participated in the polygenic risk score? A table of these SNPs should be included in the manuscript. What is "lower polygenic burden"? Does it have a threshold? Please, describe how OR

for polygenic risk score was calculated and how to interpret it.

16. In what model (dominant or recessive) OR was reported for rs7904615? What is the risk allele?

17. Do you agree that lines 167-169 can be moved to discussion from the Results

Reviewer #2 (Remarks to the Author):

Reviewer Critique

In a study, entitled "Genome-wide association study reveals a locus in ADARB2 for complete freedom from headache", Dr. Olofsson and colleagues conducted a genome-wide association study (GWAS) to assess genetic variants associated with complete freedom (protection) from headache, the authors performed a genome-wide association study of individuals who have never experienced a headache, including 63,992 individuals (2,998 individuals with complete freedom from headache and 60,994 controls) from the Danish Blood Donor Study Genomic Cohort. The authors uncovered a genome-wide significant association of an intronic variant, rs7904615[G] in ADARB2 (OR=1.20 [1.13–1.27], $p=3.92 \times 10^{-9}$) and the signal was replicated in a non-overlapping cohort from the Danish Blood Donor Study Genomic Cohort. While ADARB2 is primarily expressed in the brain, its function remains poorly understood. The authors conclude that complete freedom from headache has a genetic component, and we suggest that ADARB2 is involved in complete freedom from headache, whereas further studies are needed before ADARB2 can be proven a gene contributing to protection from headache. While these results are of potential interest to the readership of Communications Biology the authors discovery of this protective headache locus in the Danish blood donor cohort remains preliminary until association in other non-Danish cohorts is observed and/or functional studies have been conducted to confirm its role in pain signaling pathways. In this regard, it's intriguing that adenosine deaminase and adenosine signaling would have the potential to trigger headache pain by modulation of intracellular cAMP production or AMPK activity as a consequence of a change in neuronal conductance within certain brain regions. Differential gene expression studies would be the simplest approach to demonstrate upregulation of cAMP or AMPK gene activation in cell-based assays between non-carriers vs carriers of this variant.

Reviewer #3 (Remarks to the Author):

Genetic analyses for several types of headache have been performed previously, principally for migraine headache. The authors take a novel, creative approach to the genetics of headache by testing for common variant associations with complete freedom from headache (CFH), a modestly prevalent condition at only ~4%, using a Danish biobank resource. The demographics of CFH that are presented in the manuscript suggest that it may have a genetic component, especially since it has not been attributed to candidate lifestyle factors. Moreover, significant heritability has been demonstrated in twin studies. The number of cases is reasonable for this genome-wide association although not great (N=175), but the observed association is within the range expected by a power calculation. Replication supports the finding as does good behavior of the Manhattan plot.

The methods are very straightforward and the finding that reaches conventional levels of genome-wide significance and shows good evidence of replication. The results clearly add to the literature of headache genetics.

Comments

Abstract. Please add the allele frequency to the abstract. Please add specifics about the replication, at the very least “two-sided $p < 0.05$ ”. The statement about brain expression could be more specific (see below). Finally, the last two sentences could be combined and streamlined.

Starting line 181. The primary lead SNP does not replicate on the basis of $p < 0.05$ whereas other SNPs at the locus in LD with the lead SNP do. Those other SNPs reach genome-wide significance in the discovery sample and the meta-analysis is more significant than discovery alone (Table 1). This is fine and clearly supports the conclusions.

Line 186. There is a statement about meta-analysis (previous comment). However, it states that the meta-analysis is significant at $p < 0.05$. Is this an error? Was this meant to read $p < 5 \times 10^{-8}$?

Line 197. More specificity in this section about eQTLs in what is meant by “preferentially expressed”? Can this be stated in a quantitative way?

Figure 1 might zoom out in panel B or add a third panel C to show the exon/intron structure of ADARB2 and any other surrounding genes.

For Discussion. The intro mentions that in men experiencing CFH, headache could nevertheless be provoked with nitric oxide. Would the authors consider commenting on this apparent contradiction in the Discussion, perhaps in the context of their genetic finding, e.g. is ADARB2 related in any way to nitric oxide signaling?

Reply to reviewers

Reviewer #1 (Remarks to the Author):

Authors describe their discovery of a protective genetic impact on headache using genome wide association study and the Danish Blood Donor Study Genomic Cohort data.

This is an interesting finding observed in the big sample size of Danish population and then replicated in a different cohort although in the Danish population as well.

However here are questions and recommendations that can clarify and improve the manuscript.

1. The title of the paper should reflect usage of the Danish Blood Donor Study Genomic Cohort data.

- This has been corrected. Page 1, line 2.

“Genome-wide association study reveals a locus in *ADARB2* for complete freedom from headache in Danish Blood Donors”

2. I recommend to remove the second sentence from the abstract “Yet, 4% of the Danish population have never experienced headache”. This statement requires a reference and usually an abstract doesn’t contain cited publications.

- This has been corrected. Page 1, line 15.

3. Similarly, the sentence “ADARB2 is primarily expressed in the brain, but the function of the gene is poorly understood” can be removed from the abstract.

- This has been corrected. Page 1, line 26.

4. On the contrary, it would be valuable to include in the abstract different time frames for data collection for the discovery cohort (2015-2018) and for the replication cohort (2018-2020) if my understanding is correct.

- Time frames for data collection have been updated and included in the abstract. Page 1, line 20 and line 25.

5. I recommend to add information to the Introduction that 4% prevalence of headache free participants was previously published by the same group of authors. Additionally, ratio for complete freedom from headache (CFH) for sex, socio-economic status was also self-cited. Please,

tell readers if protective genetic impact was reported by other researchers since other references in the Introduction describe genetic findings associated with headaches.

- We have clarified our own publications in the introduction. Page 2, line 37-52.

We have included a sentence describing that no other studies have examined the genetics of complete freedom from headache or headache protection. Page 2, line 52.

6. Why were participants contacted twice: in 2015-2018 and then in May 2020 – August 2020? Were you able to contact all 63,992 participants in 2020?

- The DBDS is a prospective cohort where participants are continually included, and some leave the cohort. Therefore, not all 63,992 participants were recontacted in 2020. We have added a more detailed description of data collection for the cohort in the methods section. Page 2, line 65 and Page 3, line 76-81.

7. Lines 76-77 tell us that genotyping and imputation was described in the paper submitted for publication in 2018. Please, clarify if you performed QC (and how) for participants from 2018-2020 years.

- We have clarified this in the paper, Page 4, line 100-103. In short, genotype calling, quality control and imputation was done simultaneously for the entire cohort, following the same method as described in the 2018 paper.

8. Why did you include 10 PCs in the regression models?

We included 10 PC to correct for unseen biases. As seen by the scree plot (see below) most of the variance was included in PC 1-10. Page 4, line 122 and supplementary Figure 1.

A scree plot can be interesting to see (maybe in the supplemental materials).

- We have added a scree plot in supplemental materials. Page 4, line 122 and supplementary Figure 1.

What was your rationale to include sex, age, and age squared as co-variables in the regression models?

- As pain is highly influenced by sex and age we included these co-variables in the regression models. We included age squared due to the non-normal distribution of age in the DBDS Genomic Cohort. This has been added to the manuscript, Page 4, line 120-123.

A summary of base models for discovery and replication cohorts (coefficients and p-values) should be included in the main text or supplemental materials.

- It is unclear to us what is referred to with “summary of base model”. In the methods section we describe the models used in the discovery, replication, and meta-analysis (“Discovery association analysis” Page 4, line 117-128. “Replication association analysis” Page 5, line 138-

142) and “Meta-analysis” Page 5, line 145-149.) and we described the use of co-variables – which resembles the general approach for GWA. Further, we present coefficients and p-values for the lead SNP in the results section (Page 6, line 197-200) and results from the discovery, replication and meta-analysis are presented in table 2, Page 8 with coefficients and p-values, including p-values of heterogeneity in the meta-analysis.

9. Demographic description of participants (sex, age) has to be reported for both cohorts.
- **This has been added to the manuscript. Page 3, Table 1.**

10. What formula or software was used for power calculation? Results for Power calculation should be moved to the Results section.

- **We have added the software used for power calculation to the method section (Page 4, line 131-132) and moved the power calculation to the results section (Page 6, line 200-203).**

11. Please make a reference for 23andMe data, line 137.

- **We have included a reference that describes the 23andme cohort used in the migraine meta-GWAS published by Hautakangas et al., however, we would like to clarify that data from 23andme was not included in the current study, as data-sharing by 23andme does not practice open data-sharing of summary statistics as common practice within genetic research community. Page 5, line 169**

12. Please, explain differences in population prevalence for migraine (15%) and sample prevalence (8%). Please, provide references, line 140.

- **Thank you for pointing this out, as indeed LDSC demands different input values. The population prevalence is the world-wide prevalence estimated by Stovner et al. The sample prevalence refers to the case-control ratio in the summary stats used, hence refers to the ratio of the meta-GWAS published by Hautakangas et al. without samples from 23andMe and the Danish cohort. These inputs are used by LDSC to estimate the risk score on a liability scale, reflecting the expected population prevalence of 15%. This is described on the wiki of LDSC-wiki <https://github.com/bulik/ldsc/wiki/FAQ#ld-score-estimation> and in the original publication <https://doi.org/10.1101/014241>. We have updated the text to make this clear and added the references to the text. Page 6, line 171-174.**

13. Please, put the reference for LDpred2 next to it, line 145. References at the end of sentences (30, 31 - line 146) are confusing. If you want to be consistent and keep references at the end of sentences, then split the first sentence in the section about Polygenic risk score (lines 144-146).

- **This has been corrected. Page 6, line 179.**

14. Why only 5 PCs were used for the polygenic risk score regression model?

- **Thank you for pointing this out. We indeed used ‘only’ five PCs in the PRS model as we want to ensure enough power in our models for PRS. The first five PCs explain most of the variance, and**

therefore we do believe it is not necessary to include more. (See supplementary Figure 1, Scree plot)

15. How many SNPs participated in the polygenic risk score? A table of these SNPs should be included in the manuscript. What is "lower polygenic burden"? Does it have a threshold? Please, describe how OR for polygenic risk score was calculated and how to interpret it.

- As described by the developer of LDpred2, the principle of this approach is to estimate the combined effect of all common genetic variants, while correcting for linkage-disequilibrium (LD) of all variants. Using default settings, the PRS LD is derived from the 1Million + HapMap3 variants. Therefore, it does not make sense to provide a list of variants included in the PRS-calculations.

We apologized for using the somewhat arbitrary term "lower polygenic burden". This has been changed to "lower polygenic risk" and refers to individuals having a relative lower cumulative genetic risk, PRS. We acknowledge that this will be relative within the study assessed, and therefore no threshold would make sense.

16. In what model (dominant or recessive) OR was reported for rs7904615? What is the risk allele?

- On page 4, line 118 in the Method section with the sub-heading "Discovery association analysis" we state that we assume an additive genetic model. So, the OR for rs7904615 is neither dominant nor recessive, but additive.

As stated on Page 6, line 199 in the Results section with the sub-heading "Association analysis", the risk allele is [G]. This is also stated in table 2, page 8 under "Allele (effect/other)", as the risk allele is usually termed the effect allele.

17. Do you agree that lines 167-169 can be moved to discussion from the Results

- Yes, we agree and have moved the section to discussion (Page 10, line 279-282).

Reviewer #2 (Remarks to the Author):

Reviewer Critique

In a study, entitled "Genome-wide association study reveals a locus in ADARB2 for complete freedom from headache", Dr. Olofsson and colleagues conducted a genome-wide association study (GWAS) to assess genetic variants associated with complete freedom (protection) from headache, the authors performed a genome-wide association study of individuals who have never experienced a headache, including 63,992 individuals (2,998 individuals with complete freedom from headache and 60,994 controls) from the Danish Blood Donor Study Genomic Cohort. The authors uncovered a genome-wide significant association of an intronic variant, rs7904615[G] in ADARB2 (OR=1.20 [1.13–1.27], $p=3.92 \times 10^{-9}$) and the signal was replicated in a non-overlapping cohort from the Danish Blood Donor Study Genomic Cohort. While ADARB2 is primarily expressed in the brain, its function remains poorly understood. The authors conclude that complete freedom from headache has a genetic component, and we suggest that ADARB2 is involved in complete

freedom from headache, whereas further studies are needed before ADARB2 can be proven a gene contributing to protection from headache.

1. While these results are of potential interest to the readership of Communications Biology the authors discovery of this protective headache locus in the Danish blood donor cohort remains preliminary until association in other non-Danish cohorts is observed and/or functional studies have been conducted to confirm its role in pain signaling pathways.

- We agree that replication in independent cohorts, as well as conducting functional analyses, would increase the generalizability of our findings and that this study is explorative with a new phenotype. In terms of replication, we have been in contact with multiple collaborators and biobanks to encourage the use of the same or similar questions for phenotyping. Unfortunately, these efforts have not been successful. However, we have since the submission of our study, obtained additional genotyping data which has more than triple the replication cohort from 3,395 (175 individuals with CFH and 3,220 controls) to 13,032 (539 individuals with CFH and 12,493 controls). While independent replication would be important for generalizability, the significance remains after we increased the replication cohort sample. Regarding functional studies, they are indeed interesting, but are unfortunately not within the scope of this study and are not general practice for GWAS. We can only encourage research groups with the means and funds for conducting biological investigations to assess the biological mechanism of ADARB2.

2. In this regard, it's intriguing that adenosine deaminase and adenosine signaling would have the potential to trigger headache pain by modulation of intracellular cAMP production or AMPK activity as a consequence of a change in neuronal conductance within certain brain regions. Differential gene expression studies would be the simplest approach to demonstrate upregulation of cAMP or AMPK gene activation in cell-based assays between non-carriers vs carriers of this variant.

- Thank you for the comment and yes, such a connection would indeed be very interesting. We have conducted a thorough literature search but were unable to find evidence that connects ADARB2 and the nitric oxide pathway through cAMP activation. ADARB2, or Adenosine Deaminase RNA Specific B2, is a member of the adenosine deaminase family of proteins that act on RNA, not on adenosine itself. These enzymes are responsible for editing pre-mRNA molecules by deaminating specific adenosine bases to inosine within double-stranded RNA regions. This editing can lead to changes in the coding sequence of the mRNA, which can, in turn, lead to the production of different protein isoforms. It is clear from the literature that adenosine deaminases that act on adenosine, are described as being part of the cAMP signaling pathway. We anticipate that the suggestion might arise as highly similarly named ADRB2, encoding the beta-2 adrenergic receptor, has been described as direct involved in cAMP signaling. Could you please provide a reference or further details to help us understand a connection?

Reviewer #3 (Remarks to the Author):

Genetic analyses for several types of headache have been performed previously, principally for

migraine headache. The authors take a novel, creative approach to the genetics of headache by testing for common variant associations with complete freedom from headache (CFH), a modestly prevalent condition at only ~4%, using a Danish biobank resource. The demographics of CFH that are presented in the manuscript suggest that it may have a genetic component, especially since it has not been attributed to candidate lifestyle factors. Moreover, significant heritability has been demonstrated in twin studies. The number of cases is reasonable for this genome-wide association although not great (N=175), but the observed association is within the range expected by a power calculation. Replication supports the finding as does good behavior of the Manhattan plot.

The methods are very straightforward and the finding that reaches conventional levels of genome-wide significance and shows good evidence of replication. The results clearly add to the literature of headache genetics.

Comments

Abstract.

1. Please add the allele frequency to the abstract.

- This has been corrected. Page 1, line 22)

2. Please add specifics about the replication, at the very least “two-sided $p < 0.05$ ”.

- The specifics about the replication have been added to the abstract. Page 1, line 25.

3. The statement about brain expression could be more specific (see below).

- We have removed the statement of brain expression from the abstract (page 1, line 26) and specified it in the result section (page 9, line 241-247).

4. Finally, the last two sentences could be combined and streamlined.

- We have rephrased in an effort to streamline the statements. Page 1, line 29-31, it now reads:

“The genomic locus was specific for complete freedom from headache and was not associated with any primary headache disorders. Further studies are needed before *ADARB2* can be proven as a gene contributing to protection from headache.”

Starting line 181. The primary lead SNP does not replicate on the basis of $p < 0.05$ whereas other SNPs at the locus in LD with the lead SNP do. Those other SNPs reach genome-wide significance in the discovery sample and the meta-analysis is more significant than discovery alone (Table 1). This is fine and clearly supports the conclusions.

- Thank you for the comment.

5. Line 186. There is a statement about meta-analysis (previous comment). However, it states that the meta-analysis is significant at $p < 0.05$. Is this an error? Was this meant to read $p < 5 \times 10^{-8}$?

- Thank you for picking out this error in presentation, we apologize for the confusion. The p-value presented refers to marginal significance per variant tested, rather than genome-wide significance, and the text has been updated accordingly. We have also noted in the text that all of the variants are genome-wide significant in the meta-analysis, indicating robust associations across the combined cohorts, whilst having non-significant p-values of heterogeneity, suggesting consistent effect sizes and a lack of between-cohort variability, page 8, line 227-231. We now write:

"In the meta-analysis of the discovery and replication cohorts, all variants were found to be genome-wide significant with no significant heterogeneity between the cohorts. This indicates robust associations with *ADARB2* across the combined cohorts (Table 2)"

6. Line 197. More specificity in this section about eQTLs in what is meant by "preferentially expressed"? Can this be stated in a quantitative way?

- We have specified the preferentially expression and given quantitative measures, page 9, line 241-249.

" RNA expression of *ADARB2* showed tissue enrichment in the brain with 2-8 times increased expression compared to the highest expression in non-brain tissue from The Human Protein Atlas[26]. The highest expression were found in the spinal cord and the midbrain with nTPM (normalized protein-coding transcripts per million) of 31.9 and 20.1. RNA single cell type specificity showed enrichment in inhibitory neurons and oligodendrocyte precursor cells with a tau specificity score of 0.87 in the Human Protein Atlas[26]."

7. Figure 1 might zoom out in panel B or add a third panel C to show the exon/intron structure of *ADARB2* and any other surrounding genes.

- Figure 1 has been updated with a third panel C that shows exon/intron structure of *ADARB2* and surrounding genes. Page 8, Figure 1 C.

For Discussion.

The intro mentions that in men experiencing CFH, headache could nevertheless be provoked with nitric oxide. Would the authors consider commenting on this apparent contradiction in the Discussion, perhaps in the context of their genetic finding, e.g. is *ADARB2* related in any way to nitric oxide signaling?

- Thank you for your comment. Please also see comment on the putative mix-up of *ADARB2* and *ADAB2*. Still, we have added a section on nitric oxide and *ADARB2* in the discussion, page 11, line 298-301.

"We speculate if *ADARB2* could affect susceptibility to headache by decreasing the individual headache threshold. This notion is supported by our clinical study where men with CFH experienced headache when provoked with nitric oxide, a strong headache trigger."

Reviewers' comments:

Reviewer #1 (Remarks to the Author):

Authors describe their discovery of a protective genetic impact (G allele, SNP rs7904615 in ADARB2 gene) on headache using genome wide association study in the Danish Blood Donor Study Genomic Cohort data.

This is an interesting finding observed in the big sample size of Danish population and then replicated in a different cohort although in the Danish population as well.

Further epidemiological studies need to replicate the reported association between ADARB2 gene and a headache-free phenotype.

Reviewer #3 (Remarks to the Author):

Olofsson et al. have been responsive to the reviewers to the extent possible with available data. The finding is supported by the internal replication. There does not seem to be an opportunity for external replication. Nevertheless, the statistics appear robust. The presentation of findings could be still improved by a small amount of additional analysis and context. Specifically.

1. Perhaps I missed it, but could the authors provide imputation quality statistics for their lead SNPs?

2. Would the authors evaluate an interaction with sex at their lead locus? They may be unlikely to find anything significant since their number of cases is small enough that substantial sex-interaction effects might have been expected to limit discovery. Nevertheless, sex is an important determinant of headache. Could they provide a sex-stratified demographic table for cases and controls, as well? Including a test of association between case status and sex would be informative as would a test of the genetic interaction at their top locus with sex.

3. The genomic region of their candidate gene (ADARB2) is sparse in strong genome-wide significant associations with other phenotypes. For example, there are no other reported pain associations. However, there are associations with kidney function and age of menarche. Perhaps, the authors could include these observations and any further thoughts these observations might suggest about the mechanism of the association.

Reviewer #1 (Remarks to the Author):

Authors describe their discovery of a protective genetic impact (G allele, SNP rs7904615 in ADARB2 gene) on headache using genome wide association study in the Danish Blood Donor Study Genomic Cohort data. This is an interesting finding observed in the big sample size of Danish population and then replicated in a different cohort although in the Danish population as well.

Further epidemiological studies need to replicate the reported association between ADARB2 gene and a headache-free phenotype.

Thanks for the comment, and we agree that future study need to replicate the findings.

Reviewer #3 (Remarks to the Author):

Olofsson et al. have been responsive to the reviewers to the extent possible with available data. The finding is supported by the internal replication. There does not seem to be an opportunity for external replication. Nevertheless, the statistics appear robust. The presentation of findings could be still improved by a small amount of additional analysis and context.

- We agree, it is unfortunate that external replication is not possible.

Specifically.

1. Perhaps I missed it, but could the authors provide imputation quality statistics for their lead SNPs?

- We are sorry for the missing information. We have added imputation quality statistics for all genome-wide SNPs to Table 2 (page 8).

2. Would the authors evaluate an interaction with sex at their lead locus? They may be unlikely to find anything significant since their number of cases is small enough that substantial sex-interaction effects might have been expected to limit discovery. Nevertheless, sex is an important determinant of headache. Could they provide a sex-stratified demographic table for cases and controls, as well? Including a test of association between case status and sex would be informative. A test of the genetic interaction at their top locus with sex.

- We have added a sex-stratified demographics to Table 1, page 3, and assessed the distribution of case status and sex. We have furthermore added a sentence in the methods: *"There were more men among CFH cases than controls, with OR=1.99[1.85 - 2.15], p=2.44x10⁻⁷¹"*, page 3 line 78-79. Additionally, we have added data on case-status and age, sex-stratified, as a Supplementary table 1.

- Indeed, never having a headache is more prevalent in males compared with females (34% females among cases in the discovery cohort). We have carefully discussed sex-stratified and sex-interaction analyses. We conclude that the dataset does not have the necessary statistical power for further assessments. As for an interaction between a SNP and sex, we fail to state hypothesis based on the current evidence of never having a headache. Given the review requests we did performed a post-hoc analysis to explore whether the effect of our lead SNP interacted with sex: SNP * Sex. The interaction had no significant effect P-value = 0.95 nor did inclusion of the interaction term improve the regression model P-value = 0.95

3. The genomic region of their candidate gene (ADARB2) is sparse in strong genome-wide significant associations with other phenotypes. For example, there are no other reported pain associations. However, there are associations with kidney function and age of menarche. Perhaps, the authors could include these

observations and any further thoughts these observations might suggest about the mechanism of the association.

- Thank you for the comment. We have updated the discussion section and included other traits associated with ADARB2, reported in the GWAS Catalog. Page 10, line 268-271.

“PheWAS analyses of ADARB2 significant SNPs showed no association with other traits reported in the NHGRI-EBI GWAS Catalog[29]. In the GWAS Catalog ADARB2 has been associated with 60 different traits[29]. Among the traits with the lowest p-value were height (GCST90245848), age of menarche(GCST007078), acute myeloid lymphoma(GCST008413), onset of male puberty(GCST90012088), depression (GCST007342) and creatinine levels (GCST90019502)[29]. Additional functional assessments of ADARB2, including effects of intronic variants on gene expression, are needed.”

REVIEWERS' COMMENTS:

Reviewer #3 (Remarks to the Author):

The authors have been responsive to the reviewers' comments.

It's still not totally clear why controls aren't matched to cases on both age and sex, given an apparent abundance of sample that likely exceeds what is needed for maximal power of the main genetic effects. Nevertheless, there's no evidence of an interaction with sex in analysis provided in the rebuttal letter, and quite possibly little power to detect interaction, if any, as the authors also note in their rebuttal.